# Reduction in medical costs for cardiovascular diseases through innovative health awareness projects in a rural area in Japan

Ayako Shoji[1,2]☉*, Kennichi Kudo[3‡], Koichi Murashita[3‡], Shigeyuki Nakaji[3,4‡], Ataru Igarashi[1,5]☉

1 Department of Health Economics and Outcomes Research, Graduate School of Pharmaceutical Sciences, The University of Tokyo, Tokyo, Japan, 2 Medilead, Inc., Tokyo, Japan, 3 Center of Healthy Aging Innovation, Hirosaki University, Hirosaki, Japan, 4 Department of Social Medicine, School of Medicine, Hirosaki University, Hirosaki, Japan, 5 Public Health and Preventive Medicine, Yokohama City University School of Medicine, Yokohama, Japan

☉ These authors contributed equally to this work.
‡ KK, KM and SN also contributed equally to this work.
* mountain-ash795@nifty.com, a_shoji@medi-l.com

**Data Availability Statement:** Data cannot be shared publicly because the data include personal information. Data are available from Hirosaki University School of Medicine for researchers who

## Abstract

To promote health awareness and improve life expectancy in Hirosaki, a Japanese rural area, the Center of Healthy Aging Program (CHAP) was founded in 2013. The most important characteristic of CHAP is a personalized interview just after the checkup to discuss individual results. We evaluated the clinical and economic effects of CHAP by analyzing the cohort data of voluntary participants from annual health checkups since 2005 in the Iwaki district of Hirosaki. We calculated 10-year incidence risk scores for coronary heart diseases (CHDs) and stroke, and compared the risk-score trend before and after the start of CHAP by adjusting other risk factors using multivariate generalized linear regression analyses. We also predicted the 10-year incidences of CHDs and stroke based on the risk scores, for future scenarios of the two conditions, with and without CHAP, and compared them to their treatment costs between scenarios. The number of participants ranged between 808 and 1,167, from 2008 to 2016. The mean age (55 years) and proportion of women (60%) did not significantly change throughout the period. After adjusting for sex, age, outside temperature on the checkup date, the preparation effect of CHAP in 2012, and risk scores in the previous year, the annual increases in risk scores of CHDs and stroke were significant even after CHAP (+0.413, p <0.001; +0.169, p <0.001, respectively), but slightly less compared to before CHAP (+2.638, p <0.001; +1.155, p <0.001, respectively). Assuming the trend continued until 2021, the 10-year incidences of CHDs and stroke have decreases by 22,486 and 9,603, respectively, and treatment costs decreased by JPY 21,973 and 16,056 million, respectively. CHAP contributes to a significant decrease in the incidences of CHDs and stroke, and reduces economic burden on the local government.

meet the criteria for access to confidential data. Contact information is below: email coi_info@hirosaki-u.ac.jp.

**Funding:** This study was supported by the Center of Innovation Program launched by the Japan Science and Technology Agency (JPMJCE1302).

**Competing interests:** The potential competing interests of authors (if any) are otherwise summarized in the following sentences. AS is employee of Medilead Inc. KK is COE of Integrated Clinical Care Informatics, Inc. AI has received grants from Abbott Japan Inc., Abbvie G.K., Becton, Dickinson and Company, Creative-Ceuticals Inc., Eli Lilly Japan K.K., Gilead Sciences K.K., Intuitive Surgical G.K., Milliman Inc., Pfizer Inc., Sanofi Pasteur Inc., and Terumo Corporation, and personal fees from Astellas Pharma Inc., Chugai Pharmaceutical Co., Ltd., CSL Behring Japan Inc., FUJIFILM Corporation, Sanofi K.K., and Takeda Pharmaceutical Co., Ltd. outside the submitted work. These do not alter our adherence to PLOS ONE policies on sharing data and materials.

## Introduction

Cardiovascular diseases (CVDs) are among the major factors contributing to an increase in healthcare expenditures [1]. They can be effectively prevented by lifestyle modification, and community-based prevention programs have expanded in Western and Asian countries [2]. In general, rural areas were lagging behind the urban areas in terms of setting up prevention programs, as there are more aged and/or low-income residents in rural areas than in urban areas. This is because of the weak financial base of local governments and higher program costs [3] resulting from fewer opportunities to form cooperative agreements and partnerships to utilize facilities and staff. The prevention of CVDs in rural areas has remained inadequately addressed, causing an increase in medical and nursing care expenses. In Japan, delayed launch and expansion of programs in rural areas decrease budgets, as local governments incur health-care expenses for the retired elderly residents with higher CVD risks than their younger counterparts.

Hirosaki University (Aomori, Japan) initiated and accumulated results of an annual checkup for residents living in a predominantly rural agricultural community prefecture of Iwaki from 2005 onwards [4]. Hirosaki is the second-largest city in the Aomori Prefecture, located in northern Japan, and has the highest mortality rate since data was recorded In 2013, the university started an awareness and encouragement program—the Center of Healthy Aging Program (CHAP)—in the entire city of Hirosaki to promote health awareness and encourage healthy daily habits. The most important characteristic of the program is a person-alized interview that informs a citizen about health issues and related risks. This interview is based on individual results and occurs immediately after the annual checkup [4].

Previous studies reported the effects of population health approaches on health literacy and health conditions such as risk factors [5–7], mortality rates [6, 7], and hospitalization rates [2, 8] for CVDs. Most of the studies were based on regular or intensive face-to-face or online interviews designed to ensure that the participants remained motivated [9]. In contrast, the CHAP interview was held once a year, immediately after the annual health checkup. This low-intensity program design can contribute to a decrease in burden to participants, program administration, and implementation costs. However, the design possibly limits the prevention of CVDs. Some previous studies indicate that a nationwide Japanese health guidance program that includes health monitoring and online support to promote healthy behaviors, followed by health checkups, had an inadequate effect on lifestyle modification and showed clinically meaningful improvement [10, 11].

This study aimed to examine the effect of CHAP on prevention of stroke and coronary heart disease (CHD) as well as to estimate the impact on future medical costs associated with the diseases.

## Materials and methods

### Study population

We obtained data from the Iwaki cohort, which has accumulated results of an annual general health checkup since 2005 from voluntary participants before and after the start of CHAP [4]. This project is a community-based health promotion program that aims to encourage healthy lifestyle and improve the average life expectancy in Hirosaki, covering the Iwaki district of the city (Fig 1A). The Iwaki cohort data contains 3000 individual-based health-related items col-lected from both healthy and diseased individuals in the same manner every year, which allowed us to compare the trends in health parameters between participants with and without CHAP and determine its effect (Fig 1B). Even participant-reported information such as

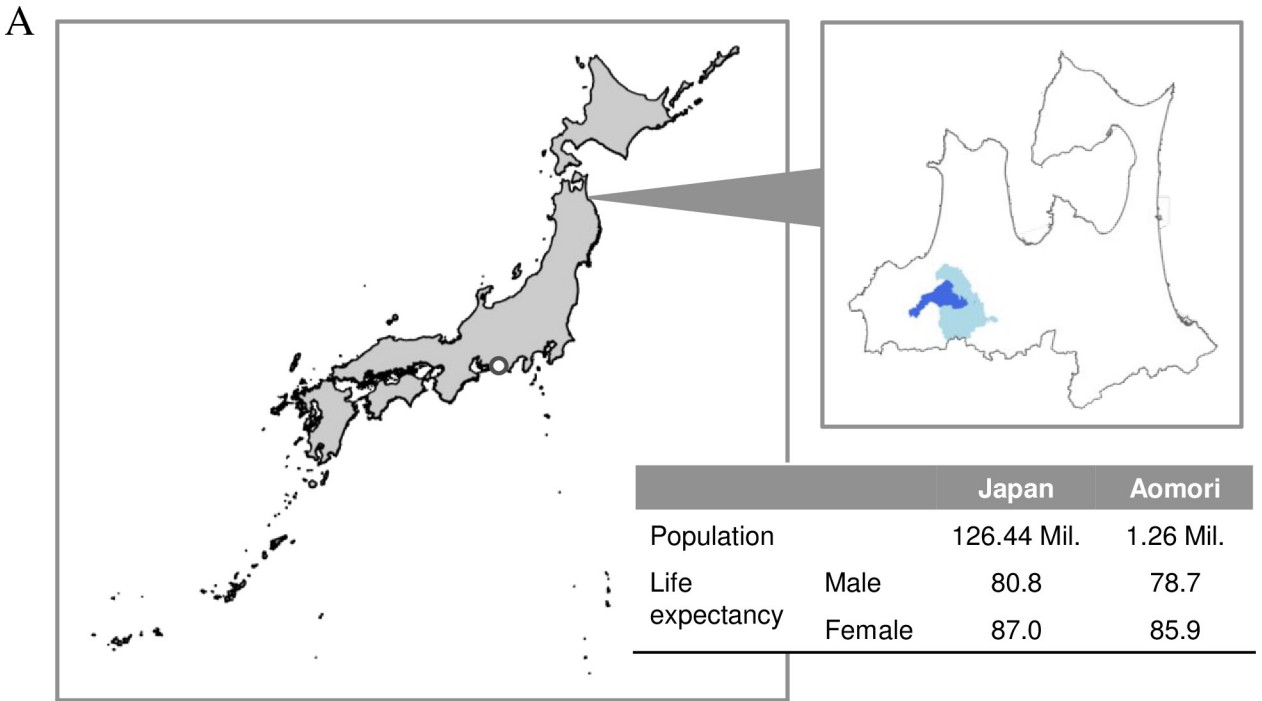

| | | Japan | Aomori |
|---|---|---|---|
| Population | | 126.44 Mil. | 1.26 Mil. |
| Life expectancy | Male | 80.8 | 78.7 |
| | Female | 87.0 | 85.9 |

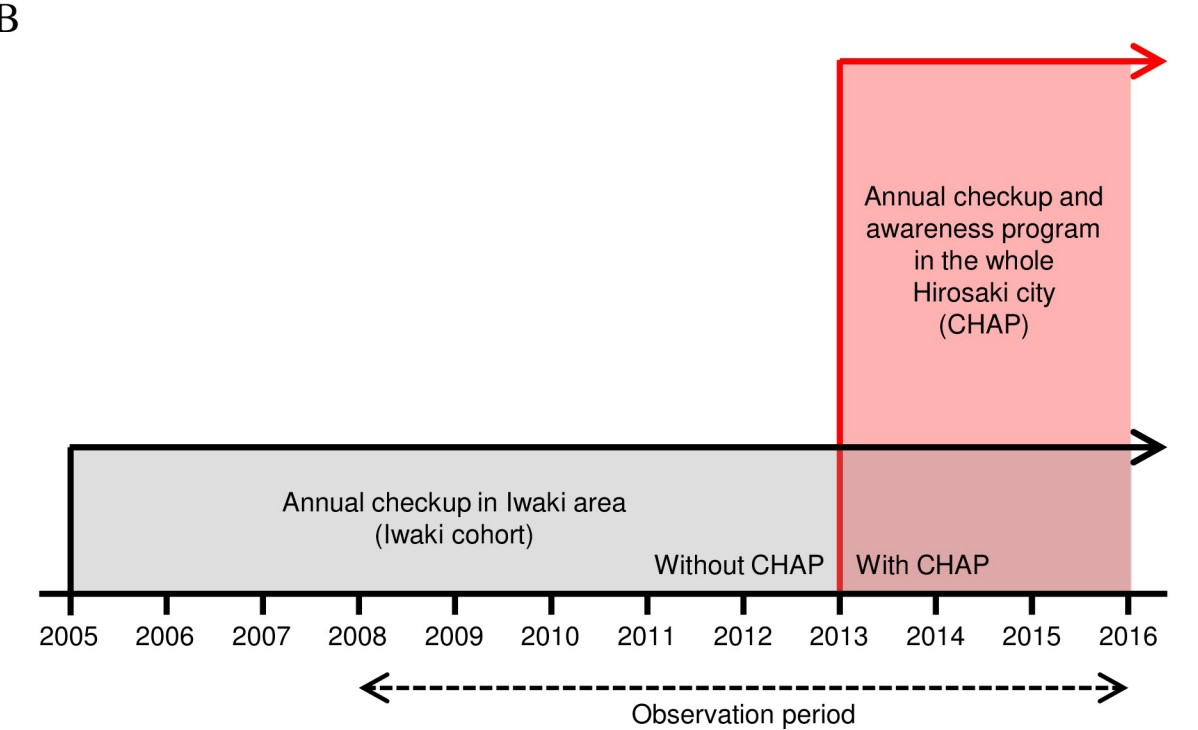

**Fig 1.** (A) Location of Aomori Prefecture, Hirosaki City (light blue), and Iwaki District (blue). (B) The Period and Coverage of Iwaki Cohort Data. Iwaki district is located in Aomori Prefecture, and Iwaki cohort data contains checkup results before and after the start of CHAP. CHAP = Center of Healthy Aging Program.

lifestyle habits was also collected by using the same questionnaire items. However, this cohort data does not contain the information of health resource utilization to be used for estimating health care costs. Some of the participants were recruited from mass advertisements while some were recommended by public health nurses. All participants provided written informed consent. The cohort study was conducted in accordance with the 1964 Helsinki Declaration, and its later amendments or comparable ethical standards, and with the approval of the Ethics Committee of Hirosaki University Graduate School of Medicine (2008–025, 2009–015, 2010–020, 2011–033, 2012–050, 2013–062, 2014–014, 2014–377, and 2016–028). This study was registered in the University Hospital Medical Information Network (UMIN-CTR, https://www.umin.ac.jp) prior to the analyses (UMIN ID: UMIN000040459). The protocol of the Iwaki cohort study has been presented [4].

We used data collected after 2008, when the proportion of participants reached 10% of residents aged 20 years or older in Iwaki, and until 2019 (the observation period; Fig 1B).

## Risk scores

We calculated risk scores of CHDs and stroke that predict the 10-year incidence risks of those diseases. Values were calculated via two steps. Firstly, risk factors of individuals enrolled in the Iwaki cohort were extracted. Details of those factors are mentioned below. Secondly, risk scores of CHDs and stroke were calculated based on equations available from other articles [12, 13]. The equation for calculating CHD risk scores was obtained from the Japanese guideline for the prevention of atherosclerotic CHDs [12], and that of stroke was done from the JPHC study, a large-scale cohort study in Japan [13] (details are described in the Supplementary Information). Both equations were derived from the results of cohort studies that are not related to the Iwaki cohort. The totalized scores were assigned for each participant considering the following risk factors for (1) CHDs: age categories in years ($\leq$44, 45–54, 55–64, 65–69, or $\geq$70), sex, smoking status, HDL cholesterol categories (<40, 40–59, or $\geq$60 mg/dL), LDL cholesterol categories (<100, 100–139, 140–159, 160–179, or $\geq$180 mg/dL), systolic and diastolic blood pressure categories (<120 and <80, 120–129 and/or 80–84, 130–139 and/or 85–89, 140–159 and/or 90–99, or $\geq$160 and/or $\geq$100, respectively), the presence of impaired glucose tolerance, and the presence of family history of early onset CHDs; and risk factors for (2) stroke: age categories ($\leq$ 44, 45–49, 50–54, 55–59, 60–64, or $\geq$65 years old), sex, systolic and diastolic blood pressure (<120 and <80, 120–129 and/or 80–84, 130–139 and/or 85–89, 140–159 and/or 90–99, or $\geq$180 and/or $\geq$110, respectively), the prescription of anti-hypertensive drugs, smoking status, body mass index (BMI) categories (<25, 25–30, or $\geq$30), and the presence of diabetes (with any anti-diabetic drugs or HbA1C of 6.5% or over). Participants who lacked checkup results of risk factors for CHDs and stroke were not excluded. We assumed no risk and counted a score of 0 for every missing risk factor. We confirmed the validity of this assumption via a sensitivity analysis in which we excluded participants with at least one missing data in the risk factors. Detail information is shown in S1 Appendix.

## Endpoints

The primary endpoints were the 10-year incidences of CHDs and stroke that were estimated based on the distribution of risk scores of both diseases; they were compared with and without CHAP. We defined the period without CHAP as 2010 to 2012, and we excluded the period from 2005 to 2007, because of the relatively low number of participants. The nationwide health guidance program start period (2008 to 2009) was also excluded from the analysis. The validity of our exclusions was confirmed using the period from 2008 to 2012 for sensitivity analysis. The period with CHAP was set from 2012 to the end of the observation period.

The secondary endpoints were the total medical costs for the estimated 10-year incidences of CHDs and stroke for patients with and without CHAP. In addition, we described the changes in both the risk scores and major risk factors throughout the observation period.

We predicted the incidences of CHDs and stroke, estimated their related medical costs, and compared them for scenarios with and without CHAP to assess their economic implications.

## Statistical analysis

We conducted multivariate analysis to establish models that explain the distribution of risk scores based on the demographic and clinical characteristics of the participants. Using generalized linear models and individual levels checkup data with calculated risk scores, we analyzed coefficients and evaluated factors associated with risk scores for each period (with and without CHAP), using the risk score as an explained variable. The explanatory variables were sex, age at the start of the period, years elapsed after the start of the period, and outside temperature at 7 a.m. on the date of health checkup; these were considered known factors of elevation in blood pressure. We could not use a generalized mixed analysis and adjust random effects among individual participants, because more than 50% of participants had received checkups only once or twice during the nine years of the observation period. However, we added an individual-specific factor and the risk score in the previous year as an explanatory variable. Participants who did not receive the checkup for two successive years were excluded. We also incorporated the preparation of CHAP in 2012 as a dummy variable in the model to provide an encouragement effect due to the expansion of information about CHAP for residents. We evaluated the trend of risk scores by using two models for periods before and after the CHAP initiation separately. To confirm the effect of the CHAP initiation on the trend of risk scores, we also used a single before and after model considering an interaction of the trend and the period. CHAP was intended to change overall attitudes and behaviors of participants, and it has been offered all participants since 2013; therefore, we did not directly compare the periods using one model.

We adapted individual participants' data in 2012 to both models and predicted individual risk scores in 2021 based on the assumption that conditions both before and after CHAP remained for 11 years. The parameters and coefficients of environmental temperature on the checkup date and the preparation of CHAP were excluded from the prediction. Based on the individual predicted risk scores, we estimated the proportion of individuals per risk score and age group. We then multiplied the proportion by the number of residents per sex and age group in Aomori [14] and incidence rates according to risk scores for estimating the incidences of CHDs and stroke within 10 years after 2022. It was assumed that the distribution of risk scores is the same throughout the prefecture. Finally, we predicted the total medical costs of the treatment per patient for CHDs or stroke by sex and age group by multiplying the incidences by the costs. The costs were calculated using national medical care expenditures per disease in 2017 [15] and the number of patients per disease throughout the country in 2017 [16].

Results were considered significant when p-values were less than 0.01. Two-sided tests were used in all comparisons and the statistical analysis software R (version 4.1.0) (The R Foundation for Statistical Computing, Vienna, Austria). was used for analysis.

## Results

### Patients' characteristics

Changes in the distribution of demographic characteristics and risk factors throughout the observation period are shown in Table 1. The number of participants increased since 2008.

**Table 1. The trend of patient characteristics, risk factors, and risk scores.**

| | | Changes within the observation period | | | | | | | | | Trend test | | | |
|---|---|---|---|---|---|---|---|---|---|---|---|---|---|---|
| Year | | 2008 | 2009 | 2010 | 2011 | 2012 | 2013 | 2014 | 2015 | 2016 | ≤ 2012 | | ≥ 2012 | |
| | | | | | | | | | | | Z | p | Z | p |
| N | | 886 | 833 | 929 | 808 | 1016 | 1054 | 1167 | 1113 | 1148 | | | | |
| Sex (female) | n (%) | 561 (63.3%) | 506 (60.7%) | 595 (64.0%) | 501 (62.0%) | 631 (62.1%) | 647 (61.4%) | 724 (62.0%) | 682 (61.3%) | 693 (60.4%) | 0.065 | 0.799 | 0.606 | 0.436 |
| Age (years) | Mean (SD) | 56.97 (13.10) | 57.28 (13.35) | 57.77 (13.68) | 57.27 (14.16) | 55.49 (15.46) | 54.39 (16.04) | 54.53 (15.43) | 54.52 (15.23) | 54.45 (15.67) | −1.191 | 0.234 | −1.565 | 0.118 |
| BMI | Mean | 23.03 (3.11) | 23.25 (3.23) | 23.08 (3.27) | 23.13 (3.25) | 22.93 (3.28) | 22.75 (3.43) | 22.80 (3.36) | 22.90 (3.49) | 22.93 (3.39) | −1.302 | 0.193 | 0.518 | 0.605 |
| | NA | 1 | 4 | 0 | 5 | 9 | 7 | 11 | 8 | 8 | | | | |
| SBP | Mean | 125.09 (19.35) | 125.58 (17.60) | 127.18 (20.29) | 133.25 (19.26) | 130.07 (19.04) | 128.28 (18.98) | 130.18 (19.93) | 122.36 (17.44) | 124.63 (18.11) | 8.199 | <0.001 | −9.188 | <0.001 |
| | NA | 0 | 2 | 3 | 1 | 0 | 0 | 1 | 0 | 0 | | | | |
| DBP | Mean | 72.33 (11.79) | 73.10 (11.68) | 72.07 (12.56) | 76.84 (12.22) | 76.18 (12.31) | 75.25 (11.69) | 77.95 (11.21) | 74.85 (11.67) | 75.04 (12.01) | 8.719 | < 0.001 | −3.154 | 0.002 |
| | NA | 0 | 2 | 3 | 1 | 0 | 0 | 1 | 0 | 0 | | | | |
| LDL-C | Mean | 117.71 (27.81) | 117.93 (26.82) | 116.07 (28.13) | 116.18 (28.48) | 119.92 (30.08) | 113.01 (28.34) | 115.36 (29.13) | 117.43 (28.36) | 117.45 (29.81) | 0.827 | 0.408 | −0.222 | 0.824 |
| | NA | 1 | 1 | 0 | 1 | 1 | 4 | 2 | 0 | 0 | | | | |
| HDL-C | Mean | 60.86 (15.02) | 62.34 (15.47) | 64.22 (16.07) | 65.36 (16.47) | 65.53 (17.43) | 64.14 (16.39) | 65.21 (16.97) | 66.85 (17.36) | 64.60 (17.14) | 6.613 | < 0.001 | 0.395 | 0.693 |
| | NA | 60 | 1 | 0 | 1 | 1 | 4 | 2 | 0 | 0 | | | | |
| HbA1C | Mean | 5.47 (0.75) | 5.59 (0.79) | 5.54 (0.72) | 5.74 (0.77) | 5.79 (0.73) | 5.87 (0.66) | 5.85 (0.65) | 5.35 (0.64) | 5.88 (0.66) | 13.363 | < 0.001 | −9.591 | < 0.001 |
| | NA | 0 | 0 | 0 | 0 | 1 | 4 | 3 | 0 | 0 | | | | |
| Smoker | n | 157 (17.7%) | 149 (17.9%) | 168 (18.1%) | 121 (15.0%) | 155 (15.3%) | 169 (16.0%) | 199 (17.1%) | 188 (16.9%) | 201 (17.5%) | 3.797 | 0.051 | 2.097 | 0.148 |
| | NA | 0 | 0 | 0 | 3 | 3 | 4 | 2 | 2 | 0 | | | | |
| Drinker | n | 361 (40.7%) | 353 (42.4%) | 387 (41.7%) | 355 (43.9%) | 407 (40.1%) | 451 (42.8%) | 494 (42.3%) | 483 (43.4%) | 502 (43.7%) | 0.005 | 0.944 | 2.491 | 0.114 |
| | NA | 2 | 4 | 0 | 2 | 4 | 2 | 2 | 2 | 0 | | | | |
| Anti-hyper-tension drug users | n | 241 (27.2%) | 215 (25.8%) | 272 (29.3%) | 237 (29.3%) | 268 (26.4%) | 282 (26.8%) | 295 (25.3%) | 310 (27.9%) | 315 (27.4%) | 0.068 | 0.794 | 0.600 | 0.439 |
| | NA | 0 | 0 | 0 | 0 | 0 | 0 | 0 | 0 | 0 | | | | |
| Anti-lipid drug user | n (%) | 104 (11.7%) | 90 (10.8%) | 105 (11.3%) | 98 (12.1%) | 121 (11.9%) | 133 (12.6%) | 144 (12.3%) | 150 (13.5) | 127 (11.1%) | 0.231 | 0.631 | 0.098 | 0.755 |
| | NA | 0 | 0 | 0 | 0 | 0 | 0 | 0 | 0 | 0 | | | | |
| Anti-diabetic drug user | n (%) | 36 (4.1%) | 54 (6.5%) | 52 (5.6%) | 42 (5.2%) | 45 (4.4%) | 49 (4.6%) | 58 (5.0%) | 54 (4.9%) | 54 (4.7%) | 0.081 | 0.776 | 0.123 | 0.726 |
| | NA | 0 | 0 | 0 | 0 | 0 | 0 | 0 | 0 | 0 | | | | |

BMI = body mass index; DBP = diastolic blood pressure; HbA1C = hemoglobin A1C; HDL-C = high density lipoprotein cholesterol; LDL-C = low density lipoprotein cholesterol; SBP = systolic blood pressure.

NA shows the number of missing records in each risk factor.

The mean systolic and diastolic blood pressure and HbA1C levels significantly increased before 2012, and decreased from 2012 onward. The mean of LDL-cholesterol remained stable both before 2012 and after 2012, and that of HDL-cholesterol increased significantly before 2012 and remained unchanged after 2012. The proportions of participants prescribed anti-hypertensive, anti-lipid, and anti-diabetic drugs remained stable both before 2012 and from 2012 onward. The mean age, which is an important risk factor of CHDs and stroke, did not

show any significant change in either period but decreased by about 3 years around 2012 (from 2011 to 2013: z = -3.361, p < 0.001). Even after the adjustment for age, the trends in mean systolic and diastolic blood pressure and HbA1C levels remained significant as described above. The proportions of participants with lifestyle habits associated with risks of CHDs and stroke, such as smoking and alcohol drinking, did not change during either period. Risk scores of CHDs significantly decreased both before 2012 and from 2012 onward (Fig 2A), and those of stroke remained stable before 2012 but significantly decreased from 2012 onward (Fig 2B).

### Factors related to changes in risk scores

Multivariate analysis showed that risk scores of both CHDs and stroke significantly increased according to years elapsed since the start of each period, (2010 for the scenario without CHAP and 2012 for that with CHAP). However, the increase per year was slightly lower from 2012 onward than it was before 2012. Higher risk scores were significantly associated with older age, men, and higher risk scores from the previous year. When CHAP started in 2012, risk scores were significantly lower for both CHDs and stroke (Table 2A and 2B). A sensitivity analysis showed that results were similar even after removing all participants with at least one missing record (see S2 Appendix). Another sensitivity analysis was conducted for the period from 2008 to 2012, defined as being without CHAP. It confirmed that derived estimated coefficients were similar to base case analysis, which showed more significant increase in risk factors than sensitivity analysis (Table 2C). A before and after model showed the decrease in risk scores of both CHDs and stroke associated with the interaction between years elapsed and the CHAP initiation, which suggested that the CHAP initiation contributed to significant decreases in risk scores of both CHDs and stroke (Table 2D).

### Predicted distribution of risk scores in 2021

Based on models of the yearly change in risk scores, we predicted the distribution of risk scores in 2021 for the two scenarios (without and with CHAP). Under the hypothetical scenario without CHAP (Figs 3B and 4B, respectively), the proportions of people with low-risk scores for CHDs and stroke were gradually decreased since 2012 (Figs 3A and 4A, respectively), while they were remained at the same level under the actual scenario with CHAP (Figs 3C and 4C, respectively).

### Estimated number of incidences and medical costs within 10 years after 2021

The 10-year incidences of CHDs and stroke in Aomori Prefecture without and with CHAP are estimated at 28,946 and 6,460 and at 19,688 and 10,085, respectively, and the reduction thereof as a result of the start of CHAP is estimated at −222,486 and −29,603 for CHDs and stroke, respectively (Table 3). This corresponded with a medical cost reduction of JPY −21,973 and −16,056 million, respectively (Table 3).

### Discussion

We revealed that CHAP improved health conditions, leading to a reduction in medical costs for CHDs and stroke. Assuming that the trend observed from 2012 onwards with CHAP continues, the incidences in the 10 years after 2020 would substantially decrease compared to the same assumption but without CHAP. Assuming yearly decreases of 2,249 and 930 (as tenth parts of predicted numbers over ten years), such figures represent about 12% and 4% of the total number of patients with CHDs and stroke under treatment, reported as 18,000 and

A

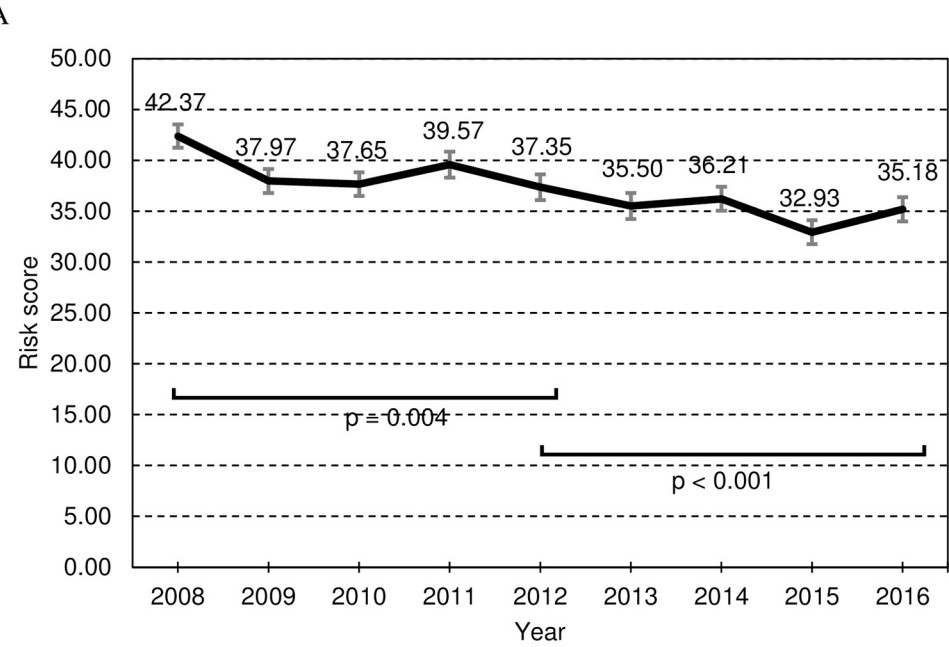

B

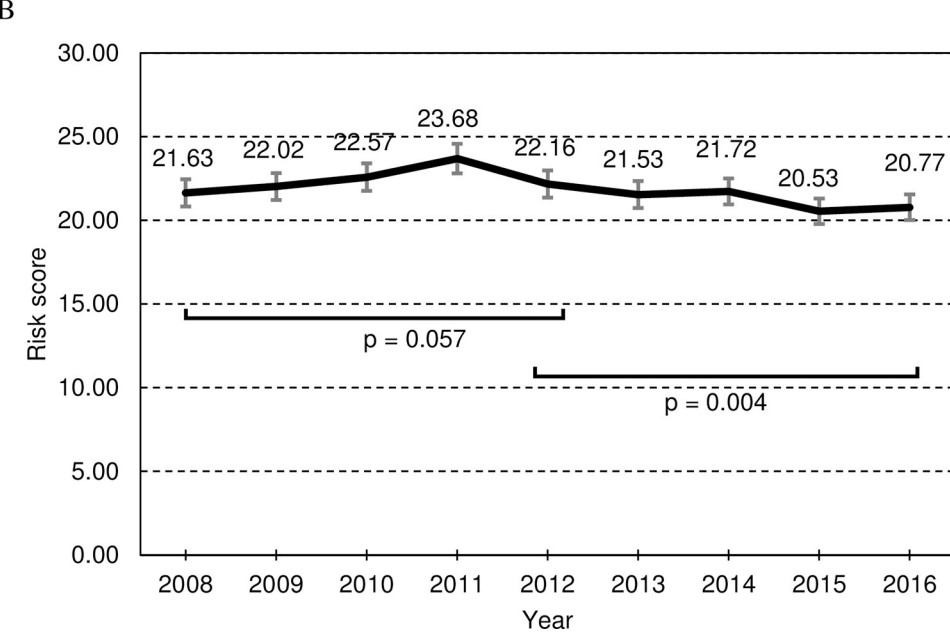

**Fig 2. The trend of risk scores.** Error bars indicate 95% confidential intervals. (A) CHDs and (B) Stroke.

25,000 in the government statistics of Aomori Prefecture in 2017 [16]. The consequent cost reduction was estimated to be more than JPY 38 billion in 10 years, which accounts for 14% of the total government medical expenditures in 2018 in Aomori Prefecture [17], that is, JPY 268 billion.

**Table 2. Factors related to risk scores.**

**(A) CHDs**

|  | Without CHAP | | | | With CHAP | | | |
|---|---|---|---|---|---|---|---|---|
|  | Est. | S.E. | t | p | Est. | S.E. | t | p |
| (Intercept) | 0.188 | 0.760 | 0.247 | 0.805 | 0.163 | 0.559 | 0.291 | 0.771 |
| Elapsed years | 2.638 | 0.324 | 8.141 | <0.001 | 0.413 | 0.077 | 5.397 | <0.001 |
| Sex (female) | −2.025 | 0.300 | 6.756 | <0.001 | −2.568 | 0.232 | 11.091 | <0.001 |
| Age at the start of each period (years) | 0.222 | 0.019 | 11.719 | <0.001 | 0.253 | 0.014 | 18.087 | <0.001 |
| Temperature on the checkup date (degree) | 0.106 | 0.064 | 1.655 | 0.098 | −0.265 | 0.043 | 6.210 | <0.001 |
| CHAP Preparation | −3.976 | 0.555 | 7.163 | <0.001 |  |  |  |  |
| Risk score of previous year | 0.767 | 0.015 | 51.881 | <0.001 | 0.739 | 0.011 | 69.061 | <0.001 |

**(B) Stroke**

|  | Without CHAP | | | | With CHAP | | | |
|---|---|---|---|---|---|---|---|---|
|  | Est. | S.E. | t | P | Est. | S.E. | t | p |
| (Intercept) | −1.414 | 0.678 | 2.086 | 0.037 | −0.687 | 0.459 | 1.499 | 0.134 |
| Elapsed years | 1.155 | 0.289 | 3.997 | <0.001 | 0.169 | 0.064 | 2.652 | 0.008 |
| Sex (female) | −2.148 | 0.275 | 7.812 | <0.001 | −2.407 | 0.202 | 11.921 | <0.001 |
| Age at the start of each period (years) | 0.202 | 0.015 | 13.219 | <0.001 | 0.195 | 0.010 | 19.188 | <0.001 |
| Temperature on the checkup date (degree) | 0.055 | 0.057 | 0.958 | 0.338 | −0.073 | 0.035 | 2.070 | 0.0385 |
| CHAP preparation | −1.959 | 0.493 | 3.975 | <0.001 |  |  |  |  |
| Risk score of previous year | 0.735 | 0.017 | 44.076 | <0.001 | 0.728 | 0.012 | 62.372 | <0.001 |

**(C) Sensitivity Analysis (from 2008 to 2012)**

|  | CHDs | | | | Stroke | | | |
|---|---|---|---|---|---|---|---|---|
|  | Est. | S.E. | t | P | Est. | S.E. | t | p |
| (Intercept) | −2.230 | 0.639 | 3.492 | <0.001 | −1.941 | 0.547 | 3.549 | <0.001 |
| Elapsed years | 2.070 | 0.130 | 15.922 | <0.001 | 0.711 | 0.110 | 6.456 | <0.001 |
| Sex (female) | −2.438 | 0.247 | 9.873 | <0.001 | −2.115 | 0.215 | 9.850 | <0.001 |
| Age at the start of each period (years) | 0.243 | 0.016 | 15.548 | <0.001 | 0.195 | 0.012 | 16.232 | <0.001 |
| Temperature on the checkup date (degree) | 0.072 | 0.047 | 1.523 | 0.128 | 0.018 | 0.040 | 0.464 | 0.642 |
| CHAP preparation | −3.309 | 0.386 | 8.574 | <0.001 | −1.322 | 0.323 | 4.095 | <0.001 |
| Risk score of previous year | 0.732 | 0.012 | 59.518 | <0.001 | 0.735 | 0.013 | 56.893 | <0.001 |

**(D) Before and after model**

|  | CHDs | | | | Stroke | | | |
|---|---|---|---|---|---|---|---|---|
|  | Est. | S.E. | t | P | Est. | S.E. | t | p |
| (Intercept) | −0.038 | 0.527 | 0.071 | 0.943 | −0.751 | 0.435 | 1.726 | 0.084 |
| Elapsed years | 2.355 | 0.349 | 6.745 | <0.001 | 1.043 | 0.292 | 3.576 | <0.001 |
| With CHAP | 0.039 | 0.397 | 0.098 | 0.922 | −0.015 | 0.331 | 0.044 | 0.965 |
| Interaction: Elapsed years and with CHAP | −1.991 | 0.358 | 5.555 | <0.001 | −0.885 | 0.300 | 2.955 | 0.003 |
| Sex (female) | −2.457 | 0.198 | 12.422 | <0.001 | −2.352 | 0.173 | 13.619 | <0.001 |
| Age at the start of each period (years) | 0.249 | 0.012 | 20.662 | <0.001 | 0.196 | 0.009 | 21.969 | <0.001 |
| Temperature on the checkup date (degree) | −0.169 | 0.036 | 4.704 | <0.001 | −0.045 | 0.030 | 1.507 | 0.132 |
| Risk score of previous year | 0.739 | 0.009 | 79.857 | <0.001 | 0.730 | 0.010 | 72.047 | <0.001 |

Est. = estimate.

The novelty of this study is that it suggested the impact of a personalized interview based on individual results on decreasing CVD risks as well as reducing medical costs. CHAP is characterized by an annual checkup with comprehensive health check items and a single interview to provide information on possible lifestyle changes to solve problems immediately after the checkup. This is different from general population health approaches that usually are intense

A

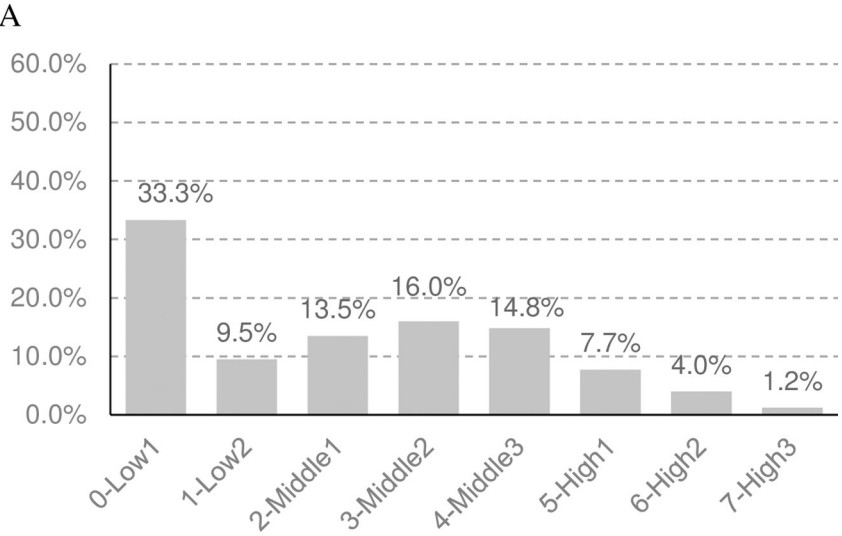

B

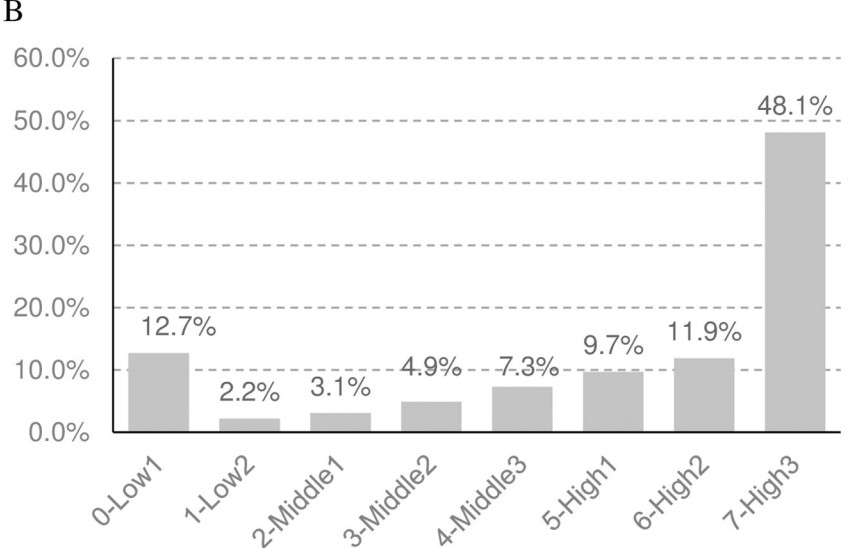

C

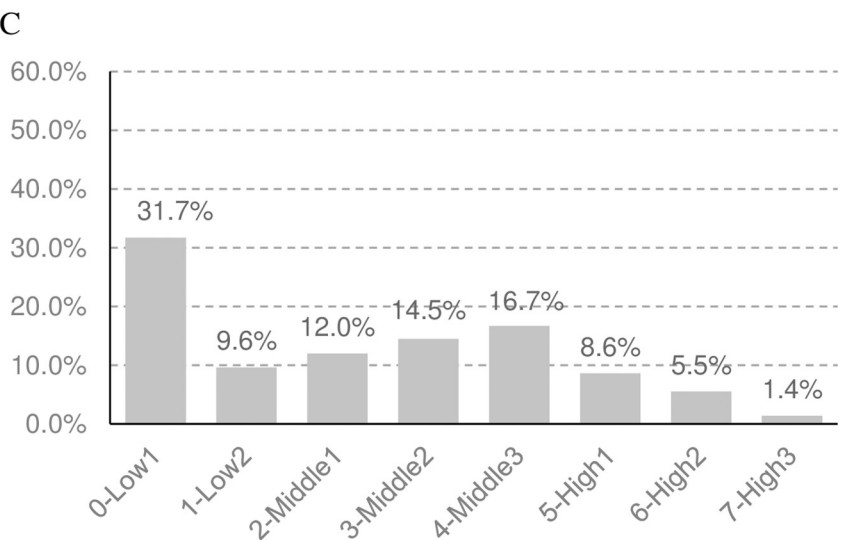

**Fig 3.** The Predicted distribution of risk scores of CHDs (A) in 2012, (B) Predicted for 2021 in the Scenario without CHAP, and (C) with CHAP. Low1, ≤35; Low2, 36–40; Middle1, 41–45; Middle2, 46–50; Middle3, 51–55; High1, 56–60; High2, 61–65; High3, ≥66.

during a specific period or consist of regular education programs [9]. A low-intensity program similar to CHAP is already offered. The Japanese nationwide health guidance program for life-style modification consists of an initial interview based on the results of health checkups, followed by monitoring of healthy behavior and continuous support for three or more interviews by instructors. However, it is offered only for high-risk persons a few months after the check-ups and its longitudinal clinical effect is unlikely to be clear [10, 11, 18]. Results from the present study suggest that even one personalized effort to raise awareness and encouragement in all participants in a year can prevent a deterioration in the laboratory parameters and an occurrence of CVDs. We believe that the consultation done immediately after the checkup effectively inspires participants to modify their behavior. We also considered the encouragement offered by volunteer checkup staff recruited from among residents. Some reports demonstrated that programs with community health workers and neighborhoods showed significant improvements in self-care behaviors [19]. CHAP can decrease future medical costs and has two implications for Aomori Prefecture: 1) a promising policy option for a more significant medical cost reduction with a smaller investment, and 2) financial leeway to implement additional healthcare measures. We expect a similar impact in other communities with the same challenges.

The effect of CHAP was remarkable for systolic blood pressure, diastolic blood pressure, and HbA1C, but not for other risk factors or lifestyle factors. Major risk factors for CHDs and stroke include cholesterol, blood pressure, diabetes mellitus, obesity, and smoking habits [20, 21]. If all these factors show improvement, a greater amount of cost reduction would be expected; however, the additional burden would require more intensive intervention. In future studies, we will assess the real medical and care resources used for each participant of CHAP rather than the aggregated medical costs reported in governmental statistics. In addition, there is a need to confirm the association between the improvement in risk factors and future economic burden. The evaluations are expected to contribute to the planning of the sophisticated application of CHAP throughout the country.

## Limitations

We believe that our estimation contained some uncertainties. We used medical costs per patient by referring to government statistics on national medical expenditure based on healthcare resource utilization for the management of stroke or CHDs, because the Iwaki cohort data does not include any data on healthcare resource utilization. However, diseases that cause individual resource use have not been sufficiently investigated and variations among patients associated with disease stages and severity were not considered in the analysis. Risk scores, which are key parameters for estimating the number of incidents in 10 years, were derived using algorithms based on cohort studies in other areas. Future analysis using individual-based exhaustive healthcare records, including medical treatments, nursing care, and other information such as the National Claims Database, are valuable for a more sophisticated evaluation of the effectiveness of CHAP. Few participants received checkups every year. In rural areas, the effect of encouragement is likely to be higher than in urban areas, owing to closer social networks. However, we expect to confirm the impact of our findings using more accumulated data in the future. To discuss the cost-effectiveness and the budget impact of CHAP, yearly management costs of CHAP must be included. By using the individual-based exhaustive

A

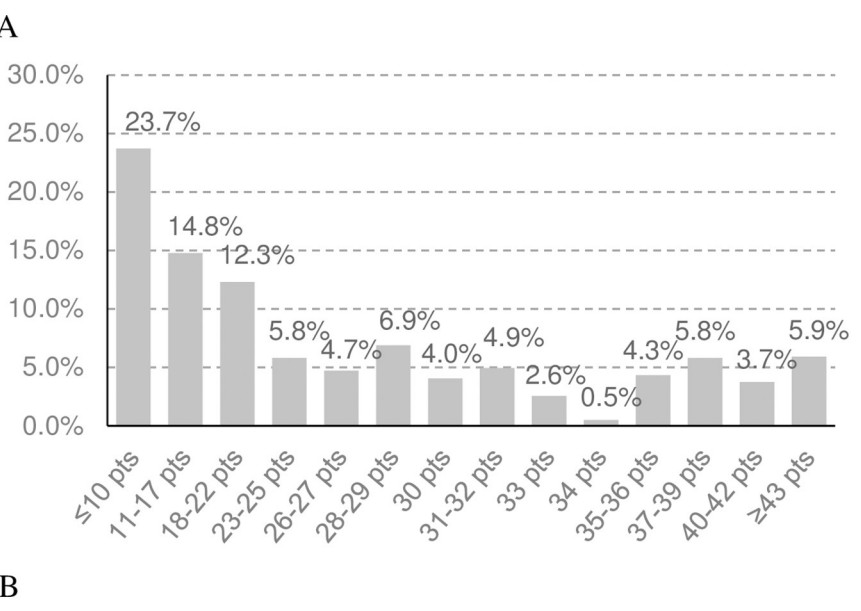

B

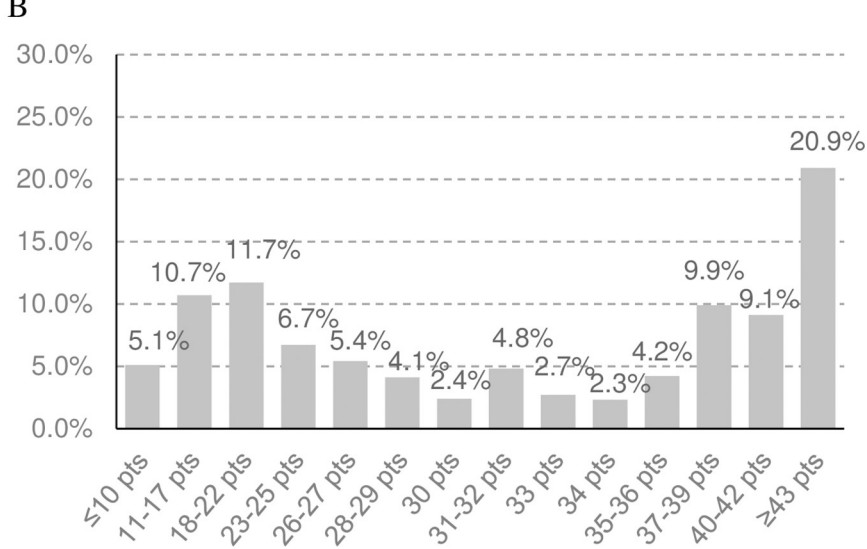

C

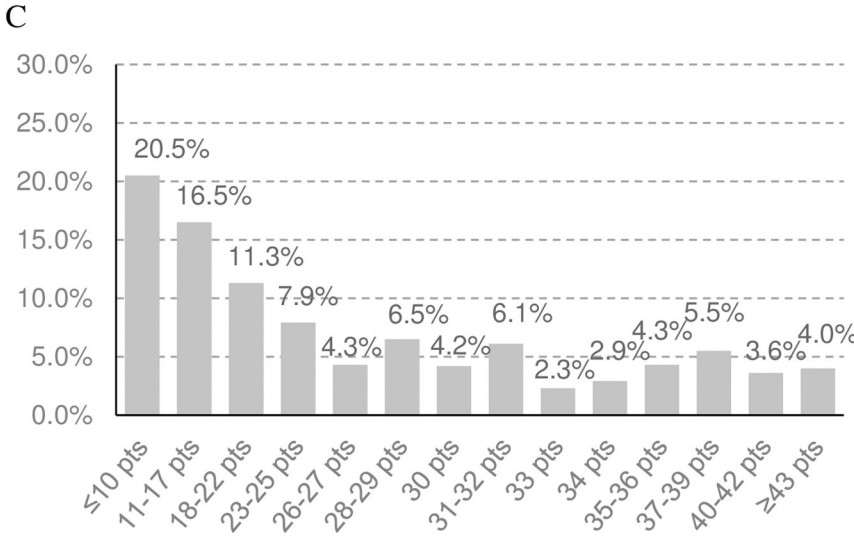

**Fig 4.** The Predicted distribution of risk scores of stroke (A) in 2012, (B) Predicted for 2021 in the Scenario without CHAP, and (C) with CHAP.

**Table 3. Estimated number of incidences and medical costs in 10 Years after 2021.**

|  | CHDs | | Stroke | |
|---|---|---|---|---|
|  | Number of incidences | Medical costs (JPY 1 M) | Number of incidences | Medical costs (JPY 1 M) |
| Without CHAP | 28,946 | 28,064 | 19,688 | 32,753 |
| With CHAP | 6,460 | 6,091 | 10,085 | 16,697 |
|  | Δ22,486 | Δ21,973 | Δ9,603 | Δ16,056 |

data, we also expect to directly compare between the periods before and after the initiation of CHAP as well as between participants of CHAP and non-participants.

## Conclusions

We found that an innovative project for awareness and encouragement in the rural area has the potential for cost reduction in the treatment of CVDs and corresponding reduction in the economic burden on the local government.

## Supporting information

**S1 Appendix. Risk score to estimate the 10-year incidence risk of CHDs and stroke.**
(DOCX)

**S2 Appendix. Factors related to changes in risk scores after the removal of cases with missing records.**
(DOCX)

## Acknowledgments

We gratefully acknowledge the contribution of the residents of the Iwaki area of Hirosaki, who voluntarily participated in the community health examination. We also thank the universities, governmental institutions, local and prefectural governments, and private companies that constitute the Center of Healthy Aging Program for their continued collaboration and support.

We would like to thank Editage (www.editage.jp) for English language editing.

## Author Contributions

**Conceptualization:** Ayako Shoji, Kennichi Kudo, Koichi Murashita, Shigeyuki Nakaji, Ataru Igarashi.

**Data curation:** Ayako Shoji.

**Formal analysis:** Ayako Shoji.

**Investigation:** Ayako Shoji.

**Methodology:** Ayako Shoji, Kennichi Kudo, Koichi Murashita, Shigeyuki Nakaji, Ataru Igarashi.

**Supervision:** Ataru Igarashi.

**Visualization:** Ayako Shoji.

**Writing – original draft:** Ayako Shoji, Ataru Igarashi.

**Writing – review & editing:** Ayako Shoji, Kennichi Kudo, Koichi Murashita, Shigeyuki Nakaji, Ataru Igarashi.

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
