## [Decision Letter · Decision Letter 0]

5 Apr 2022

PONE-D-22-04944Reduction in medical costs for cardiovascular diseases through innovative health awareness projects in a rural area in JapanPLOS ONE

Dear Dr. Shoji,

Thank you for submitting your manuscript to PLOS ONE. After careful consideration, we feel that it has merit but does not fully meet PLOS ONE’s publication criteria as it currently stands. Therefore, we invite you to submit a revised version of the manuscript that addresses the points raised during the review process.

We look forward to receiving your revised manuscript.

Kind regards,

Yoshihiro Fukumoto

Academic Editor

PLOS ONE

Journal Requirements:

"The potential competing interests of authors (if any) are otherwise summarized in the following sentences. AS is employee of Medilead Inc. KK is COE of Integrated Clinical Care Informatics, Inc. AI has received grants from Abbott Japan Inc., Abbvie G.K., Becton, Dickinson and Company, Creative-Ceuticals Inc., Eli Lilly Japan K.K., Gilead Sciences K.K., Intuitive Surgical G.K., Milliman Inc., Pfizer Inc., Sanofi Pasteur Inc., and Terumo Corporation, and personal fees from Astellas Pharma Inc., Chugai Pharmaceutical Co., Ltd., CSL Behring Japan Inc., FUJIFILM Corporation, Sanofi K.K., and Takeda Pharmaceutical Co., Ltd. outside the submitted work."

Reviewers' comments:

Reviewer's Responses to Questions

**Comments to the Author**

1. Is the manuscript technically sound, and do the data support the conclusions?

Reviewer #1: Partly

Reviewer #2: Yes

2. Has the statistical analysis been performed appropriately and rigorously? 

Reviewer #1: Yes

Reviewer #2: Yes

3. Have the authors made all data underlying the findings in their manuscript fully available?

Reviewer #1: No

Reviewer #2: Yes

4. Is the manuscript presented in an intelligible fashion and written in standard English?

Reviewer #1: Yes

Reviewer #2: Yes

5. Review Comments to the Author

Reviewer #1: Overall, the manuscript is well written. However, there are several points that should be improved in terms of potential additional analyses and justifications of the proposed methodology.

1) It should be clearly stated what is the novelty of this study and what it adds to the existing literature.

2) It is not clear what kind of data were used. It is stated that data were obtained from the Iwaki cohort, so presumably the analysis was based on patient’s level data, but this is not stated. It maybe that aggregate data were used, but again, this is not stated or justified. This is somewhat explained in the limitation section, but it should be explained clearly and in greater details in the methods section.

3) The method section is not clear, a good way to explain the clinical and economic impact of the CHAP programme is a before and after econometric method. The authors should explain why this was not possible considering the data limitation.

4) The risk score to estimate the 10-year incidence risks of CHDs and stroke is referenced, but a brief explanation of what is it and how it works should be provided (perhaps as supporting information).

5) Costs are calculated by multiplying the incidence of CHDs and stroke by medical costs. The estimation is based on assumptions and not on a costing methodology. It is not clear why, if patents level data are available, costs were not estimated with, for instance a two-part model (accounting for 0 costs and skewed distribution of healthcare costs) or any other sound methodologies.

6) Table and figures could be presented in a more appealing way. E.g., in Table 1, numbers and percentages could be presented within one single cell. Similarly, mean and SD could be presented together. E.g., age in years in 2008 could be presented in the following way 56.97 (13.10). Figures 2 should be fully labelled to aid the reader.

7) In the Conclusions, it is stated that the innovative project for awareness and encouragement contributes to a significant cost reduction in the treatment of CVDs and reduces the economic burden on the local government. In reality, this is based on assumptions as a formal econometric analysis was not conducted. It could be said, however, that the innovative project has the potential for cost reduction etc. etc.

Reviewer #2: Major comments:

This study was revealed the effect of CHAP as an innovative project on the prevention of stroke and coronary heart disease as well as to contribute to a cost reduction in the treatment of these diseases. The concept of the present study was valuable but there are some concerns as described below.

1. In Table 1, the subjects appear to be younger after 2012 than before. Author described that “The mean age, which is an important risk factor of CHDs and stroke, did not show any significant change in either period but decreased by about 3 years around 2012” in page10, line 182-184.

It is questionable whether there is any statistical discrepancy in age before 2012 and after.

2. In Table 1, the author showed significant trend changes in SBP, DBP, and HbA1C, before and after 2012, but I wonder if they are adjusted for age, because the mean age seems to be younger after 2012 than before.

3. I don't understand how you could show “a more remarkable increase in risk factors (Table 2C).” in page16, line 207-208. Could you explain more about it?

4. In page 22; line 262-263, the author described that “the effect of CHAP was remarkable for SBP, DBP, and HbA1C, but not for other risk factors or lifestyle factors.”Is it possible that there was a bias in the content of lifestyle interview or guidance in CHAP? It would be better to mention standardization of instructional content.

5. In figure 3 & 4, the differences between B and C were understood, but the results for A and C seem to be very similar. Careful explanations are needed about the results for A and C.

Minor comments:

1. Page 16; line 196-197. This figure legend showed the definitions of risk groups in Figure 3. It should be in Figure 3.

6. PLOS authors have the option to publish the peer review history of their article (what does this mean?). If published, this will include your full peer review and any attached files.

Reviewer #1: **Yes: **Giorgio Ciminata

Reviewer #2: No

---

## [Author Response · Author response to Decision Letter 0]

27 May 2022

Dear Reviewers, 

Thank you for your careful reviews and constructive comments. We have revised the manuscript according to your suggestions and responded in detail in the file titled "response_to_reviewers_economic_impact_20May2022.docx". We would be grateful if you would review the updated version. 

Sincerely, 

Ayako Shoji

Reviewer #1

Comments to the Authors

Overall, the manuscript is well written. However, there are several points that should be improved in terms of potential additional analyses and justifications of the proposed methodology..

Author Response: Thank you for the valuable comments. We have updated according to your suggestions as described below.

Comment 1. It should be clearly stated what is the novelty of this study and what it adds to the existing literature.

Author Response: Thank you for the indication. We have added descriptions that explain the novelty of our study and the difference of methods and results from previous studies in Discussion section. 

Comment 2. It is not clear what kind of data were used. It is stated that data were obtained from the Iwaki cohort, so presumably the analysis was based on patient’s level data, but this is not stated. It maybe that aggregate data were used, but again, this is not stated or justified. This is somewhat explained in the limitation section, but it should be explained clearly and in greater details in the methods section.

Author Response: Thank you for the indication. We found that our explanation was insufficient as you indicated, and then have added descriptions in Materials and Method section.

Comment 3. The method section is not clear, a good way to explain the clinical and economic impact of the CHAP programme is a before and after econometric method. The authors should explain why this was not possible considering the data limitation.

Author Response: Thank you for the important suggestion. We could not use a difference-in-difference method, because the cohort data that we used does not include persons who hadn’t participate CHAP after the initiation of CHAP. Alternatively, we used generalized linear models as considering risk factors of CVDs and differences among participants, but we did not compare between the periods before and after the initiation of CHAP by using one model, because we thought that CHAP is a program, which intend to change participants’ attitude and behavior and its effect too complex to consider in one model. We have added this explanation in Materials and Method, and Limitations section. 

Comment 4. The risk score to estimate the 10-year incidence risks of CHDs and stroke is referenced, but a brief explanation of what is it and how it works should be provided (perhaps as supporting information).

Author Response: Thank you for the indication. We have added the explanation as a supporting information. 

Comment 5. Costs are calculated by multiplying the incidence of CHDs and stroke by medical costs. The estimation is based on assumptions and not on a costing methodology. It is not clear why, if patents level data are available, costs were not estimated with, for instance a two-part model (accounting for 0 costs and skewed distribution of healthcare costs) or any other sound methodologies.

Author Response: Thank you for the indication. As was answered your comment #2, we found that our explanation was unclear. In the cohort data that we used, there is no information to be used for estimating the amount of healthcare resource uses. We have added the description in Materials and Methods section and Limitation section.

Comment 6. Table and figures could be presented in a more appealing way. E.g., in Table 1, numbers and percentages could be presented within one single cell. Similarly, mean and SD could be presented together. E.g., age in years in 2008 could be presented in the following way 56.97 (13.10). Figures 2 should be fully labelled to aid the reader.

Author Response: Thank you for the significant comment. We have corrected Table 1 and Figure 2 as you indicated. 

Comment 7. In the Conclusions, it is stated that the innovative project for awareness and encouragement contributes to a significant cost reduction in the treatment of CVDs and reduces the economic burden on the local government. In reality, this is based on assumptions as a formal econometric analysis was not conducted. It could be said, however, that the innovative project has the potential for cost reduction etc. etc.

Author Response: Thank you for the important suggestion. We have changed the description as you proposed in Conclusion section.

 

Reviewer #2

Comments to the Authors

This study was revealed the effect of CHAP as an innovative project on the prevention of stroke and coronary heart disease as well as to contribute to a cost reduction in the treatment of these diseases. The concept of the present study was valuable but there are some concerns as described below.

Author Response: Thank you for the evaluation and the important comments. We have updated according to your suggestions as described below.

Comment 1. In Table 1, the subjects appear to be younger after 2012 than before. Author described that “The mean age, which is an important risk factor of CHDs and stroke, did not show any significant change in either period but decreased by about 3 years around 2012” in page10, line 182-184.

Author Response: Thank you for the important indication. We thought that your concern associated with your next comment #2 and then have added additional analyses according to your suggestion in comment #2. We are happy if you could see our response to comment #2. 

Comment 2. In Table 1, the author showed significant trend changes in SBP, DBP, and HbA1C, before and after 2012, but I wonder if they are adjusted for age, because the mean age seems to be younger after 2012 than before.

Author Response: Thank you for the significant suggestion. We additionally analyzed according to your suggestion and confirmed that the trends of SBP, DBP, and HbA1C during the period before and after 2012 did not significantly changed, even after the adjustment for age. We added the description about the results in Results section.

Comment 3. I don't understand how you could show “a more remarkable increase in risk factors (Table 2C).” in page16, line 207-208. Could you explain more about it?

Author Response: Thank you for the indication. We have changed the description to clarify.

Comment 4. In page 22; line 262-263, the author described that “the effect of CHAP was remarkable for SBP, DBP, and HbA1C, but not for other risk factors or lifestyle factors.”Is it possible that there was a bias in the content of lifestyle interview or guidance in CHAP? It would be better to mention standardization of instructional content.

Author Response: Thank you for the significant suggestion. The process of CHAP has been published as a protocol that we referred in the present study. Lifestyle information is collected by using a same questionnaire every year and then we believe that there is only a limited bias arising from interviewers. We have explained it in Materials and Method section.

Comment 5. In figure 3 & 4, the differences between B and C were understood, but the results for A and C seem to be very similar. Careful explanations are needed about the results for A and C.

Author Response: Thank you for the indication. We have changed the description about the difference between results.

Minor comment 1. Page 16; line 196-197. This figure legend showed the definitions of risk groups in Figure 3. It should be in Figure 3.

Author Response: Thank you for the indication. We have corrected.

---

## [Decision Letter · Decision Letter 1]

28 Aug 2022

PONE-D-22-04944R1Reduction in medical costs for cardiovascular diseases through innovative health awareness projects in a rural area in JapanPLOS ONE

Dear Dr. Shoji,

Thank you for submitting your manuscript to PLOS ONE. After careful consideration, we feel that it has merit but does not fully meet PLOS ONE’s publication criteria as it currently stands. Therefore, we invite you to submit a revised version of the manuscript that addresses the points raised during the review process.

The authors should respond Reviewer 1, and would be encouraged for re-submission.

We look forward to receiving your revised manuscript.

Kind regards,

Yoshihiro Fukumoto

Academic Editor

PLOS ONE

Reviewers' comments:

Reviewer's Responses to Questions

**Comments to the Author**

1. If the authors have adequately addressed your comments raised in a previous round of review and you feel that this manuscript is now acceptable for publication, you may indicate that here to bypass the “Comments to the Author” section, enter your conflict of interest statement in the “Confidential to Editor” section, and submit your "Accept" recommendation.

Reviewer #1: (No Response)

Reviewer #2: All comments have been addressed

Reviewer #3: All comments have been addressed

2. Is the manuscript technically sound, and do the data support the conclusions?

Reviewer #1: Partly

Reviewer #2: Yes

Reviewer #3: Yes

3. Has the statistical analysis been performed appropriately and rigorously? 

Reviewer #1: No

Reviewer #2: Yes

Reviewer #3: Yes

4. Have the authors made all data underlying the findings in their manuscript fully available?

Reviewer #1: Yes

Reviewer #2: Yes

Reviewer #3: Yes

5. Is the manuscript presented in an intelligible fashion and written in standard English?

Reviewer #1: Yes

Reviewer #2: Yes

Reviewer #3: Yes

6. Review Comments to the Author

Reviewer #1: Overall, most comments have been addressed, however there are still two points of major concern:

1)It was previously suggested to use a before and after econometric method to explain the clinical and economic impact of the CHAP programme. The author responded saying that “Because CHAP was intended to change overall attitudes and behaviors of participants and it has been offered all participants since 2013, therefore we did not directly compare between the periods by using one model”.

The all point of a public health intervention such as the CHAP programme is to change overall attitudes and behaviours of participants. So, it is still not clear why a single before and after econometric model was not used. Also, it is understood that “the period with CHAP was set from 2012 to the end of the observation period” and that CHAP was offered to all participants since 2013. In this case, is not clear how the trend for risk score can be evaluated in the “before” model. If a single before -after econometric model cannot be used, I would have expected to see a risk score calculation at baseline (before CHAP introduction) then a “trend for risk score” for the years following the introduction of CHAP.

It maybe that this is provided in the existing analysis, but it is not clearly presented or framed.

On a minor note, the sentence could have been constructed better (e.g. “therefore” after “because” is not correct grammar).

2)This comment refers to an issue not previously picked up but of a major concern:

On page 7 (line 133-136 of the revised copy) the following is stated: “Participants who lacked checkup results of risk factors for CHDs and stroke were not excluded. We assumed no risk and counted a score of 0 for every missing risk factor. Detail information is shown in Supporting information”. It is not clear what this means. Are there missing records (e.g., if a patient did not have a risk factor for CHDs, 0 should be reported) or simply if the risk factors were not identified they were not reported? I believe that latter applies. In this case, I would simply delete the whole sentence, somewhat misleading.

However, if the former applies, assuming that patients with missing records on risk factors have 0 risk is quite a big assumption. This “dilutes” the proportion of patients with none or little risk. If it is a complete case (but this should be stated) the missing records should be reported and discarded. Alternatively, a multiple imputation should be carried out. It is appreciated that this for whatever reason may not be feasible, but if so, it should be clearly stated. At the minimum, the number of patients with missing records should be reported, and a sensitivity analysis excluding these patients should be conducted.

Reviewer #2: All comments have been addressed. Authors replied adequately to my questions. I think the revise manuscript is much better.

Reviewer #3: After reviewing the revised version of this manuscript, I believe that authors have satisfactorily answered the queries raised by reviewers. The manuscript is well-written, has excellent-quality data and statistical analyses. I have only but one minor suggestion, that is:

In lines 36,37 of the abstract, authors state that "CHDs and stroke were significant even after CHAP (+ 0.413, p < .001; + 0.169, p < .001, respectively), but slightly more compared to before CHAP" I believe that authors meant to write "slightly less" instead of "slightly more".

I also recommend authors to perform a final check on English and grammar.

7. PLOS authors have the option to publish the peer review history of their article (what does this mean?). If published, this will include your full peer review and any attached files.

Reviewer #1: **Yes: **Giorgio Ciminata

Reviewer #2: No

Reviewer #3: No

---

## [Author Response · Author response to Decision Letter 1]

23 Sep 2022

Dear Reviewers,

Thank you for your careful reviews and constructive comments. We have revised the manuscript according to your suggestions and would be grateful if you would review the updated version. 

Sincerely, 

Ayako Shoji

---

## [Editor Report · Decision Letter 2]

1 Nov 2022

Reduction in medical costs for cardiovascular diseases through innovative health awareness projects in a rural area in Japan

PONE-D-22-04944R2

Dear Dr. Shoji,

We’re pleased to inform you that your manuscript has been judged scientifically suitable for publication and will be formally accepted for publication once it meets all outstanding technical requirements.

Kind regards,

Yoshihiro Fukumoto

Academic Editor

PLOS ONE
---

## [Editor Report · Acceptance letter]

7 Nov 2022

PONE-D-22-04944R2 

Reduction in medical costs for cardiovascular diseases through innovative health awareness projects in a rural area in Japan 

Dear Dr. Shoji:

I'm pleased to inform you that your manuscript has been deemed suitable for publication in PLOS ONE. Congratulations! Your manuscript is now with our production department. 

Kind regards, 

on behalf of

Dr. Yoshihiro Fukumoto 

Academic Editor

PLOS ONE